# Impact of the COVID-19 pandemic on timeliness and equity of measles, mumps and rubella vaccinations in North East London: a longitudinal study using electronic health records

Nicola Firman 🔾 , Milena Marszalek 🔾 , Ana Gutierrez, Kate Homer, Crystal Williams, Gill Harper, Isabel Dostal, Zaheer Ahmed, John Robson, Carol Dezateux

NF and MM contributed equally.

Wolfson Institute of Population Health, Queen Mary University of London, London, UK

**Correspondence to**
Nicola Firman;
nicola.firman@qmul.ac.uk and Milena Marszalek;
m.marszalek@qmul.ac.uk

## ABSTRACT

**Objectives** To quantify the effect of the COVID-19 pandemic on the timeliness of, and geographical and sociodemographic inequalities in, receipt of first measles, mumps and rubella (MMR) vaccination.

**Design** Longitudinal study using primary care electronic health records.

**Setting** 285 general practices in North East London.

**Participants** Children born between 23 August 2017 and 22 September 2018 (pre-pandemic cohort) or between 23 March 2019 and 1 May 2020 (pandemic cohort).

**Main outcome measure** Receipt of timely MMR vaccination between 12 and 18 months of age.

**Methods** We used logistic regression to estimate the ORs (95% CIs) of receipt of a timely vaccination adjusting for sex, deprivation, ethnic background and Clinical Commissioning Group. We plotted choropleth maps of the proportion receiving timely vaccinations.

**Results** Timely MMR receipt fell by 4.0% (95% CI: 3.4% to 4.6%) from 79.2% (78.8% to 79.6%) to 75.2% (74.7% to 75.7%) in the pre-pandemic (*n*=33 226; 51.3% boys) and pandemic (*n*=32 446; 51.4%) cohorts, respectively. After adjustment, timely vaccination was less likely in the pandemic cohort (0.79; 0.76 to 0.82), children from black (0.70; 0.65 to 0.76), mixed/other (0.77; 0.72 to 0.82) or with missing (0.77; 0.74 to 0.81) ethnic background, and more likely in girls (1.07; 1.03 to 1.11) and those from South Asian backgrounds (1.39; 1.30 to 1.48). Children living in the least deprived areas were more likely to receive a timely MMR (2.09; 1.78 to 2.46) but there was no interaction between cohorts and deprivation (Wald statistic: 3.44; p=0.49). The proportion of neighbourhoods where less than 60% of children received timely vaccination increased from 7.5% to 12.7% during the pandemic.

**Conclusions** The COVID-19 pandemic was associated with a significant fall in timely MMR receipt and increased geographical clustering of measles susceptibility in an area of historically low and inequitable MMR coverage. Immediate action is needed to avert measles outbreaks and support primary care to deliver timely and equitable vaccinations.

## STRENGTHS AND LIMITATIONS OF THIS STUDY

⇒ We used routine primary care electronic health records available in near real time for an entire population of children registered with all National Health Service general practices in one region of London.

⇒ Coding of routine childhood vaccinations by primary care teams in North East London is enabled by data entry templates with standardised coding enabling high-quality recording of childhood vaccinations at the point of care.

⇒ We used robust statistical methods to investigate inequalities in measles, mumps and rubella timeliness and the impact of the COVID-19 pandemic.

⇒ Ethnic background was not recorded in the primary care electronic health records of more than one-third of children in our study sample.

## INTRODUCTION

The COVID-19 pandemic disrupted routine healthcare and services across the UK, through rising COVID-19 infections as well as the introduction of social distancing measures and lockdowns.[1] The UK Joint Committee on Vaccination and Immunisation emphasised the importance of continued receipt of routine vaccinations throughout periods of lockdown.[2]

In the 12 months to March 2021, an average of 90.3% of children scheduled to receive a first measles, mumps and rubella (MMR) vaccination had been vaccinated by 24 months of age in England. This was approximately 0.3% lower than for the same period to March 2020, with average levels in both years well below the WHO coverage target of 95%.[3]

These national averages conceal significant geographical inequity. The most recent annual local authority coverage

data demonstrate that in London, MMR coverage fell from 83.6% to 82.4% at 24 months from 2019 to 2021.[3] These findings indicate significant disruption across routine child vaccination schedules which are worse in regions with historically low uptake,[4] reflecting pre-existing inequalities observed in MMR uptake in the UK[5] resulting in a measles outbreak and loss of measles 'elimination status' in 2019.[6]

Research investigating the impact of the COVID-19 pandemic on routine vaccination schedules found that MMR uptake during April 2020 dropped initially by 42.5% in London in comparison with the same time period during 2019.[7] In addition to reduced vaccination uptake, a recent systematic review has highlighted the consequential impacts on vaccination inequalities during pandemics,[8] identifying four studies which reported that inequalities in routine vaccination coverage worsened during the pandemic compared with pre-pandemic months. A study in Pakistan[9] identified greater reductions in routine vaccination coverage among children whose parents had lower education than children whose parents had received higher educational levels, while a Colombian study[10] found reduced vaccination coverage among children living in rural compared with urban areas. Studies taking place in the USA identified widening inequalities in vaccination coverage during the pandemic by race[11] and Medicaid enrolment.[12]

The findings from this systematic review highlight the importance of understanding the impact of the COVID-19 pandemic on MMR in the UK. Further reductions in MMR uptake will increase the risk of future measles outbreaks, particularly in London where a significant proportion of children start school without the full protection offered by MMR vaccination.[13]

Methods currently used to assess vaccine coverage lack information on timeliness, as well as social, ethnic and geographical inequalities. Because of their retrospective nature, these methods are not actionable. This is important as it has been acknowledged that regional averages conceal geographical clusters of susceptibility in smaller areas which fuel outbreaks.[14] We examined the impact of the COVID-19 pandemic on timeliness of the first MMR vaccination in North East London (NEL). Specifically, we aimed to quantify the impact of COVID-19 on timeliness of the first MMR vaccination and investigate whether inequalities in receipt of timely MMR vaccination and its geographical clustering were amplified during the pandemic. We also aimed to report the number of measles and mumps cases recorded in primary care in NEL occurring during the pandemic period and equivalent pre-pandemic period.

## METHODS
### Study design and setting
We carried out a longitudinal study using primary care electronic health records (EHRs) from 285 general

practices (GPs) in seven geographically contiguous NEL Clinical Commissioning Groups (CCGs): Barking & Dagenham, City & Hackney, Havering, Newham, Redbridge, Tower Hamlets and Waltham Forest. The study protocol can be found in online supplemental file 1 and STrengthening the Reporting of OBservational studies in Epidemiology checklist in online supplemental file 2.

### Study population
We defined two cohorts of children eligible to receive their first MMR vaccination between 12 and 18 months of age in the 19 months before and after 23 March 2020—the date at which the first national lockdown commenced in the UK. The pre-pandemic cohort comprised those born between 23 August 2017 and 22 September 2018, and the pandemic cohort those born between 23 March 2019 and 1 May 2020.

### Data sources
Pseudonymised data were provided from the NEL Discovery Data Service, which receives primary care EHR data on a daily basis from all GPs in NEL. Demographic and clinical data were extracted for 1 192 630 children born between September 2001 and October 2021, ever registered with a NEL GP and including children who may have died or left the area. Data were extracted on 23 November 2021 and included all clinical events up to 1 November 2021. All data were extracted and managed according to UK National Health Service (NHS) information governance requirements.[15]

### Data processing
We identified 519 465 children with a NEL GP registration at the time of their first birthday (online supplemental figure S1) and retained only those eligible for the pandemic and pre-pandemic cohorts (figure 1).

We extracted sociodemographic and geographical data for each child, together with—for each child—all clinical events relating to MMR procedures. Documentation of the processing of MMR events can be found in online supplemental figure S2, tables S1 and S2. Using access to calendar week, month and year of birth we derived a proxy date of birth combining the date of the first day of the week of the calendar week of birth with month and year of birth.

### Outcome of interest
We defined timely MMR vaccination as receipt of the first MMR vaccination between 12 and 18 months of age, which is consistent with NHS England's definition of a timely MMR vaccination for the Quality and Outcomes Framework (QOF).[16] A vaccination considered not timely may have been given too early (before 12 months of age), late (after 18 months of age) or not recorded (never or not yet received).

Cases of measles and mumps were identified in primary care EHRs as events with relevant Systematized

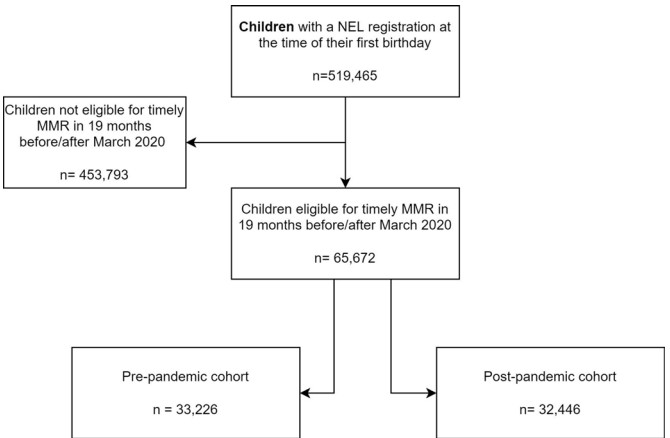

**Figure 1** Study sample. MMR, measles, mumps and rubella; NEL, North East London.

Nomenclature of Medicine (SNOMED) Clinical Terms (see online supplemental table S3). Only the first instance of each diagnosis code was retained.

## Covariates of interest

We merged 2019 Index of Multiple Deprivation (IMD) rank[17] into the datafile using the 2011 Lower Super Output Area (LSOA)—an area with an average population of 1500 people or 650 households—as the linkage field, and categorised IMD rank into the national quintiles from most to least deprived. Ethnic background was categorised using the NHS classification from information recorded in the primary care record. We grouped ethnic background into four mutually exclusive groups: white ('white British', 'white Irish' or 'any other white background'); black ('black African', 'black Caribbean' or 'any other black background'); South Asian ('Indian', 'Pakistani', 'Bangladeshi' or 'Sri Lankan'); and mixed/other ('any other ethnic background', 'mixed ethnicity', 'Chinese' or 'Asian other'). A missing category was created where ethnicity was not coded in the primary care record.

## Statistical analyses

We explored variation in the proportion of children receiving a timely MMR vaccination by cohort, sex, CCG, ethnic background and IMD quintile, and described the differences in the proportion of children receiving timely MMR vaccination in each cohort by these covariates.

For children with a GP-recorded address with an associated LSOA in NEL, we plotted choropleth maps of the proportion of children receiving timely MMR vaccination in each cohort and the change in proportion between the two cohorts by LSOA, to visualise geographical clustering of MMR vaccination timeliness. LSOAs with fewer than 10 eligible children in either the pre-pandemic or pandemic cohorts were suppressed.

We conducted binary logistic regression to estimate the adjusted odds (OR and 95% CI) of timely MMR vaccination after adjustment for covariates. We tested an interaction between cohort and IMD quintile to assess whether

COVID-19 had widened inequalities in timely vaccination. All analyses were conducted using Stata (V.MP/15.0).

## Patient and public involvement

We involved patients and the public in the communication of study results and dissemination within the local community using accepted principles from the UK Standards for Public Involvement.[18] The aim was to raise awareness of the importance of inequalities in timely childhood vaccinations. We established a patient advisory group, comprising six parents, to co-produce dissemination materials. The patient and public involvement group reflected on vaccination inequalities, the study design and how results were delivered. Participants expressed reservations about the categorisation of ethnic background and whether more granular categories could be used in future research. They discussed communication and visualisation of results. Dissemination of results is ongoing and informed by advice about accessing hard-to-reach and existing community groups.

## RESULTS

The pre-pandemic and pandemic cohorts comprised 33 226 (51.3% boys) and 32 446 (51.4% boys) children, respectively (figure 1). The cohorts were similar with respect to demographic characteristics: the majority lived in the most deprived areas and were from diverse ethnic backgrounds (table 1 and online supplemental table S4). Timely MMR receipt was 4.0% (95% CI: 3.4% to 4.6%) lower in the pandemic compared with the pre-pandemic cohort.

Children from white, mixed/other and black ethnic backgrounds had the lowest—and children from South Asian ethnic backgrounds the highest—percentage of timely MMR receipt (table 2). There was a strong positive gradient in vaccination timeliness by IMD quintile: relative to those living in the least deprived quintile, the proportion of children receiving a timely MMR vaccination was 10.8% (8.6% to 13.0%) and 14.3% lower (11.8% to 16.8%) in the pre-pandemic and pandemic cohorts, respectively.

The proportion of LSOAs where fewer than 60% of children received a timely MMR vaccination increased from 7.5% (90) to 12.7% (153) in the pandemic cohort (figure 2A,B). These were clustered in parts of City & Hackney, Newham, Redbridge, and Barking & Dagenham. Almost half of LSOAs where fewer than 60% of children received timely MMR vaccinations were assigned to the most deprived IMD quintile compared with one-third in the pre-pandemic cohort (online supplemental table S5). The proportion of children receiving a timely MMR vaccination fell during the pandemic period in 634 (52.7%) out of 1203 LSOAs (figure 3), and these were predominantly located in Tower Hamlets and City & Hackney. The proportion increased in 367 LSOAs (30.5%) and remained the same in 13 (1.1%).

**Table 1** Sample characteristics

| | Pre-pandemic cohort (n=33 226) | | | Pandemic cohort (n=32 446) | | | All (n=65 672) | | |
|---|---|---|---|---|---|---|---|---|---|
| | n | % | 95% CI | n | % | 95% CI | n | % | 95% CI |
| **Sex** | | | | | | | | | |
| Male | 17 055 | 51.3 | 50.8 to 51.9 | 16 665 | 51.4 | 50.8 to 49.2 | 33 720 | 51.4 | 51.0 to 51.7 |
| Female | 16 169 | 48.7 | 48.1 to 49.2 | 15 781 | 48.6 | 48.1 to 49.2 | 31 950 | 48.6 | 48.3 to 49.0 |
| Other | 2 | | | 0 | | | 2 | | |
| **CCG** | | | | | | | | | |
| Barking & Dagenham | 3916 | 11.8 | 11.4 to 12.1 | 3819 | 11.8 | 11.4 to 12.1 | 7735 | 11.8 | 11.5 to 12.0 |
| City & Hackney | 4771 | 14.4 | 14.0 to 14.7 | 4631 | 14.3 | 13.9 to 14.7 | 9402 | 14.3 | 14.1 to 14.6 |
| Havering | 3684 | 11.1 | 10.8 to 11.4 | 3657 | 11.3 | 10.9 to 11.6 | 7341 | 11.2 | 10.9 to 11.4 |
| Newham | 6458 | 19.4 | 19.0 to 19.9 | 6210 | 19.1 | 18.7 to 19.6 | 12 668 | 19.3 | 19.0 to 19.6 |
| Redbridge | 4971 | 15.0 | 14.6 to 15.3 | 4793 | 14.8 | 14.4 to 15.2 | 9764 | 14.9 | 14.6 to 15.1 |
| Tower Hamlets | 4605 | 13.9 | 13.5 to 14.2 | 4598 | 14.2 | 13.8 to 14.6 | 9203 | 14.0 | 13.8 to 14.3 |
| Waltham Forest | 4821 | 14.5 | 14.1 to 14.9 | 4738 | 14.6 | 14.2 to 15.0 | 9559 | 14.6 | 14.3 to 14.8 |
| **Ethnic background** | | | | | | | | | |
| White | 9579 | 28.8 | 28.3 to 29.3 | 8938 | 27.6 | 27.1 to 28.0 | 18 517 | 28.2 | 27.9 to 28.5 |
| Mixed and other | 3813 | 11.5 | 11.1 to 11.8 | 3766 | 11.6 | 11.3 to 12.0 | 7579 | 11.5 | 11.3 to 11.8 |
| South Asian | 5881 | 17.7 | 17.3 to 18.1 | 5802 | 17.9 | 17.5 to 18.3 | 11 683 | 17.8 | 17.5 to 18.1 |
| Black | 2054 | 6.2 | 5.9 to 6.4 | 1992 | 6.1 | 5.9 to 6.4 | 4046 | 6.2 | 6.0 to 6.3 |
| Missing | 11 899 | 35.8 | 35.2 to 36.3 | 11 948 | 36.8 | 36.3 to 37.4 | 23 847 | 36.3 | 35.9 to 36.7 |
| **IMD quintile** | | | | | | | | | |
| Most deprived | 12 436 | 37.4 | 36.9 to 38.0 | 11 995 | 37.0 | 36.4 to 37.5 | 24 431 | 37.2 | 36.8 to 37.6 |
| 2 | 13 464 | 40.5 | 40.0 to 41.1 | 13 306 | 41.0 | 40.5 to 41.5 | 26 770 | 40.8 | 40.4 to 41.1 |
| 3 | 4533 | 13.6 | 13.3 to 14.0 | 4400 | 13.6 | 13.2 to 13.9 | 8933 | 13.6 | 13.3 to 13.9 |
| 4 | 1883 | 5.7 | 5.4 to 5.9 | 1956 | 6.0 | 5.8 to 6.3 | 3839 | 5.9 | 5.7 to 6.0 |
| Least deprived | 847 | 2.6 | 2.4 to 2.7 | 754 | 2.3 | 2.2 to 2.5 | 1601 | 2.4 | 2.3 to 2.6 |
| Missing | 63 | 0.2 | 0.1 to 0.2 | 35 | 0.1 | 0.1 to 0.2 | 98 | 0.2 | 0.1 to 0.2 |
| **Timely MMR vaccination*** | | | | | | | | | |
| Yes | 26 315 | 79.2 | 78.8 to 79.6 | 24 402 | 75.2 | 74.7 to 75.7 | 50 717 | 77.2 | 76.9 to 77.5 |
| No | 6911 | 20.8 | 20.4 to 21.2 | 8044 | 24.8 | 24.3 to 25.3 | 14 955 | 22.8 | 22.5 to 23.1 |
| Early | 180 | 0.5 | 0.5 to 0.6 | 120 | 0.4 | 0.3 to 0.4 | 300 | 0.5 | 0.4 to 0.5 |
| Late | 1678 | 5.1 | 4.8 to 5.3 | 932 | 2.9 | 2.7 to 3.1 | 2610 | 4.0 | 3.8 to 4.1 |
| Not yet received | 5053 | 15.2 | 14.8 to 15.6 | 6992 | 21.5 | 21.1 to 22.0 | 12 045 | 18.3 | 18.0 to 18.6 |

*Receipt of first MMR vaccination between 12 and 18 months of age. Not timely is further broken down into three groups: early (before age 12 months), late (after age 18 months) and not yet received.
CCG, Clinical Commissioning Group; IMD, Index of Multiple Deprivation; MMR, measles, mumps and rubella.

After adjustment, timely MMR receipt was less likely in the pandemic cohort (0.79; 0.76 to 0.82), children from black (0.70; 0.65 to 0.76), mixed/other (0.77; 0.72 to 0.82) or with missing (0.77; 0.74 to 0.81) ethnic backgrounds, and more likely in girls (1.07; 1.03 to 1.11) and those from South Asian backgrounds (1.39; 1.30 to 1.48). Children living in the least deprived areas were more likely to receive a timely MMR (2.09; 1.78 to 2.46; Wald test statistic: 201.66; p<0.0001; figure 4 and online supplemental table S6), but there was no interaction between cohorts and deprivation (Wald statistic:

3.44; p=0.49). Relative to children registered with a GP in Newham, timely MMR receipt was less likely among children in Barking & Dagenham (0.88; 0.82 to 0.94), City & Hackney (0.67; 0.63 to 0.71) or Redbridge (0.69; 0.64 to 0.73) and more likely among those registered to a GP in Havering (1.53; 1.40 to 1.66), Tower Hamlets (1.52; 1.42 to 1.64) and Waltham Forest (1.21; 1.14 to 1.30).

In NEL, there were 20 measles and 34 mumps cases recorded in primary care during the pandemic period,

**Table 2** Proportion of children with timely MMR vaccination* in each cohort, and the percentage point difference between pre-pandemic and pandemic cohorts, by sociodemographic characteristics

| | Pre-pandemic cohort (n=26 315) | | | Pandemic cohort (n=24 402) | | | Percentage point difference† | |
|---|---|---|---|---|---|---|---|---|
| | n | % | 95% CI | n | % | 95% CI | % | 95% CI |
| **Sex** | | | | | | | | |
| Male | 13 362 | 78.3 | 77.7 to 79.0 | 12 480 | 74.9 | 74.2 to 75.5 | –3.4 | –4.3 to –2.5 |
| Female | 12 953 | 80.1 | 79.5 to 80.7 | 11 922 | 75.5 | 74.9 to 76.2 | –4.6 | –5.5 to –3.7 |
| **CCG** | | | | | | | | |
| Barking & Dagenham | 2999 | 76.6 | 75.2 to 77.9 | 2725 | 71.3 | 69.9 to 72.8 | –5.3 | –7.3 to –3.3 |
| City & Hackney | 3403 | 71.3 | 70.0 to 72.6 | 2897 | 62.6 | 61.2 to 63.9 | –8.7 | -10.6 to –6.8 |
| Havering | 3205 | 87.0 | 85.9 to 88.0 | 3095 | 84.6 | 83.4 to 85.8 | –2.4 | –4.0 to –0.8 |
| Newham | 5077 | 78.6 | 77.6 to 79.6 | 4717 | 76.0 | 74.9 to 77.0 | –2.6 | –4.1 to –1.1 |
| Redbridge | 3713 | 74.7 | 73.5 to 75.9 | 3483 | 72.7 | 71.4 to 73.9 | –2.0 | –3.7 to –0.3 |
| Tower Hamlets | 3977 | 86.4 | 85.3 to 87.3 | 3737 | 81.3 | 80.1 to 82.4 | –5.1 | –6.6 to –3.6 |
| Waltham Forest | 3941 | 81.7 | 80.6 to 82.8 | 3748 | 79.1 | 77.9 to 80.2 | –2.6 | –4.2 to –1.0 |
| **Ethnic background** | | | | | | | | |
| White | 7865 | 82.1 | 81.3 to 82.9 | 6874 | 76.9 | 76.0 to 77.8 | –5.2 | –6.4 to –4.0 |
| Mixed and other | 2872 | 75.3 | 73.9 to 76.7 | 2641 | 70.1 | 68.6 to 71.6 | –5.2 | –7.2 to –3.2 |
| South Asian | 4966 | 84.4 | 83.5 to 85.3 | 4824 | 83.1 | 82.2 to 84.1 | –1.3 | –2.6 to 0.0 |
| Black | 1517 | 73.9 | 71.9 to 75.7 | 1374 | 69.0 | 66.9 to 71.0 | –4.9 | –7.7 to –2.1 |
| Missing | 9095 | 76.4 | 75.7 to 77.2 | 8689 | 72.7 | 71.9 to 73.5 | –3.7 | –4.8 to –2.6 |
| **IMD quintile** | | | | | | | | |
| Most deprived | 9710 | 78.1 | 77.3 to 78.8 | 8779 | 73.2 | 72.4 to 74.0 | –4.9 | –6.0 to –3.8 |
| 2 | 10 523 | 78.2 | 77.5 to 78.8 | 9865 | 74.1 | 73.4 to 74.9 | –4.1 | –5.1 to –3.1 |
| 3 | 3675 | 81.1 | 79.9 to 82.2 | 3437 | 78.1 | 76.9 to 79.3 | –3.0 | –4.7 to –1.3 |
| 4 | 1628 | 86.5 | 84.8 to 87.9 | 1649 | 84.3 | 82.6 to 85.9 | –2.2 | –4.4 to 0.0 |
| Least deprived | 753 | 88.9 | 86.6 to 90.8 | 660 | 87.5 | 85.0 to 89.7 | –1.4 | –4.6 to 1.8 |
| Missing | 26 | 41.3 | 29.9 to 53.7 | 12 | 32.3 | 20.6 to 51.2 | –9.0 | –28.7 to 10.7 |

*Receipt of MMR vaccination between 12 and 18 months of age.
†Proportion receiving timely MMR vaccination in pandemic cohort minus the proportion receiving timely MMR vaccination in the pre-pandemic cohort.
CCG, Clinical Commissioning Group; IMD, Index of Multiple Deprivation; MMR, measles, mumps and rubella.

compared with 325 and 140, respectively, in the equivalent pre-pandemic period (online supplemental table S7).

## DISCUSSION
### Summary of key findings
In the period preceding the COVID-19 pandemic in NEL, only 79% of children received their first MMR vaccination on time; this proportion fell by an average of 4% during the pandemic. The gap between the most and least deprived areas increased by 3.5% during the pandemic period. While this relative inequality did not appear to worsen during the pandemic, these average figures conceal marked inequity at a lower geographical level: the percentage of LSOAs, where fewer than 60% of children received their MMR on time increased from 7.5% to

12.7% in the pandemic, particularly in the most deprived LSOAs. Hence, delayed receipt of MMR is geographically clustered in more deprived neighbourhoods, and this has worsened during the pandemic. These findings are in an area of London where no CCG has met the WHO MMR target of 95% coverage.[3] In the absence of national data, our analyses show for the first time how far this region of London is from achieving the new QOF target for MMR timeliness of 90%–95%.[16]

### Strengths and limitations
We used routine primary care EHRs available in near real time for an entire population of children registered with all NHS GPs in one region of London. We have been able to demonstrate—in a geographically contiguous area—neighbourhoods with very high proportions of children who are not immunised promptly. These results further

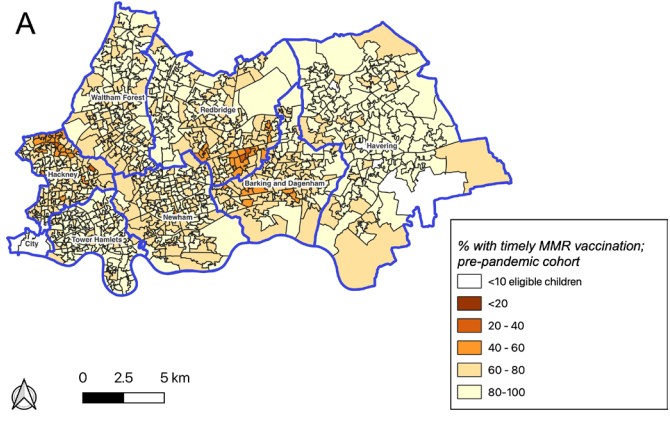

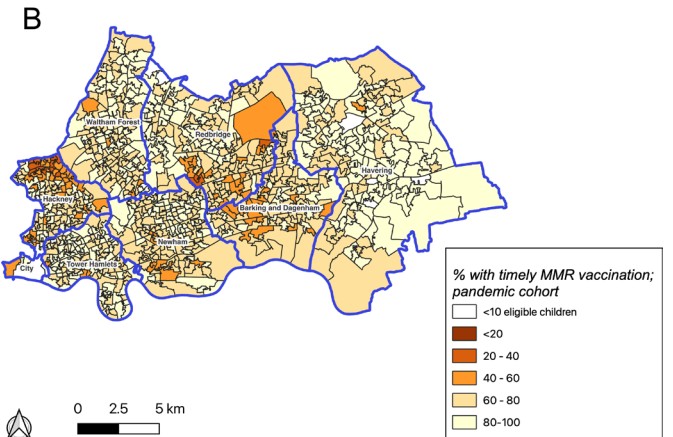

**Figure 2** (A) Proportion of children receiving timely MMR vaccination in the pre-pandemic cohort, by 2011 LSOA (B) Proportion of children receiving timely MMR vaccination in the pandemic cohort, by 2011 LSOA. MMR, measles, mumps and rubella; LSOA, lower super output area of the child's home access, as recorded in their general practice electronic health record.

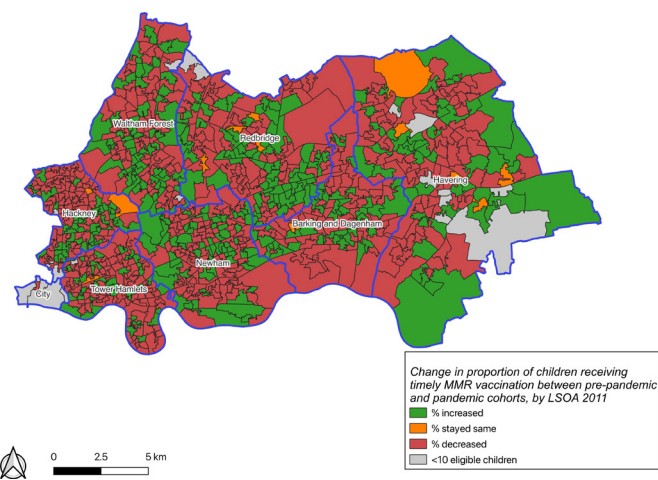

**Figure 3** Change[1] in the proportion of children receiving timely MMR vaccination between pre-pandemic and pandemic cohorts, by 2011 LSOA. [1]Proportion receiving timely MMR vaccination in pandemic cohort minus the proportion receiving timely MMR vaccination in the pre-pandemic cohort. MMR, measles, mumps and rubella; LSOA, lower super output area of the child's home address, as recorded in their general practice electronic health record.

highlight the inequalities in vaccination timeliness and infection risks experienced by poorer children, their families and communities.

Coding of routine childhood vaccinations by primary care teams in NEL is enabled by data entry templates with standardised coding enabling high-quality recording of childhood vaccinations at the point of care. We used robust statistical methods to investigate inequalities in MMR timeliness and the impact of the COVID-19 pandemic.

Limitations include missing ethnic codes in 36% of the cohort. Our analyses suggest that those with missing ethnicity were less likely to receive a timely MMR vaccine during the pandemic and highlight the importance of improving routine recording of ethnicity in primary care. While our study has focused on timeliness of the first MMR vaccination, it is important to recognise that two doses of MMR are essential for full protection.[19] Additional research investigating timeliness of the second MMR vaccination would further our understanding and

improve identification of children with increased measles susceptibility.

## Comparison with existing literature

Our findings align with trends indicating a global decrease in uptake of MMR vaccination, both in developing and developed countries.[10 20 21] Some studies reported a decline in uptake of more than 50% during the height of the first wave of the pandemic.[22–24] Globally, measles-containing vaccine coverage estimates were 7.9% lower than expected had there been no COVID-19 pandemic, affecting an estimated 8.9 million children.[25] In England, initial reductions in the number of children receiving their first MMR were followed by a short period of recovery, compared with the same period in 2019, despite continued physical distancing measures remaining in place.[26] However, this increase was short-lived, and the weekly count of children receiving their MMR vaccination in 2020 remained consistently lower or the same as in 2019 for the rest of 2020. Our findings may be explained in part by evidence from qualitative research studies demonstrating that a transition to remote consultations during the pandemic caused some confusion for parents around attending services for routine vaccination.[4] Internationally, fear of COVID-19 exposure in healthcare settings was also found to play a large role in decreased vaccination uptake,[25 27 28] despite evidence that the risk to benefit ratio was in favour of continuing vaccination delivery during the pandemic.[29]

The link between childhood vaccination inequalities and ethnicity has been explored in other studies, demonstrating evidence of reduced timeliness in certain ethnic groups.[30] However, there is heterogeneity within these results according to geographical area of interest.[31] We

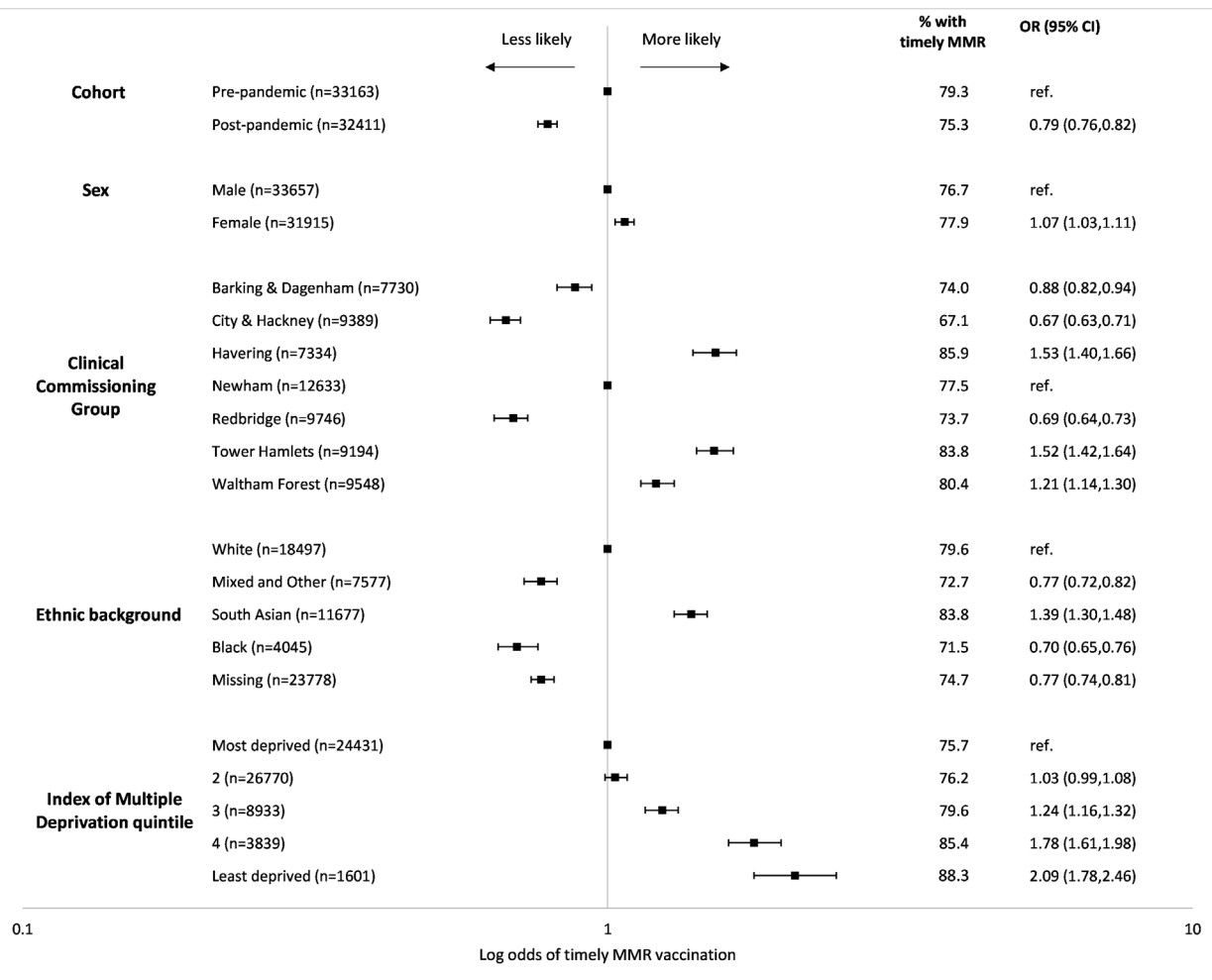

| | | Less likely ← | → More likely | % with timely MMR | OR (95% CI) |
|---|---|---|---|---|---|
| **Cohort** | Pre-pandemic (n=33163) | | | 79.3 | ref. |
| | Post-pandemic (n=32411) | | | 75.3 | 0.79 (0.76,0.82) |
| **Sex** | Male (n=33657) | | | 76.7 | ref. |
| | Female (n=31915) | | | 77.9 | 1.07 (1.03,1.11) |
| **Clinical Commissioning Group** | Barking & Dagenham (n=7730) | | | 74.0 | 0.88 (0.82,0.94) |
| | City & Hackney (n=9389) | | | 67.1 | 0.67 (0.63,0.71) |
| | Havering (n=7334) | | | 85.9 | 1.53 (1.40,1.66) |
| | Newham (n=12633) | | | 77.5 | ref. |
| | Redbridge (n=9746) | | | 73.7 | 0.69 (0.64,0.73) |
| | Tower Hamlets (n=9194) | | | 83.8 | 1.52 (1.42,1.64) |
| | Waltham Forest (n=9548) | | | 80.4 | 1.21 (1.14,1.30) |
| **Ethnic background** | White (n=18497) | | | 79.6 | ref. |
| | Mixed and Other (n=7577) | | | 72.7 | 0.77 (0.72,0.82) |
| | South Asian (n=11677) | | | 83.8 | 1.39 (1.30,1.48) |
| | Black (n=4045) | | | 71.5 | 0.70 (0.65,0.76) |
| | Missing (n=23778) | | | 74.7 | 0.77 (0.74,0.81) |
| **Index of Multiple Deprivation quintile** | Most deprived (n=24431) | | | 75.7 | ref. |
| | 2 (n=26770) | | | 76.2 | 1.03 (0.99,1.08) |
| | 3 (n=8933) | | | 79.6 | 1.24 (1.16,1.32) |
| | 4 (n=3839) | | | 85.4 | 1.78 (1.61,1.98) |
| | Least deprived (n=1601) | | | 88.3 | 2.09 (1.78,2.46) |

0.1                                    1                                    10

Log odds of timely MMR vaccination

**Figure 4** Adjusted log odds of timely MMR vaccination.[1] [1]Model mutually adjusting for cohort, sex, Clinical Commissioning Group, ethnic background and Index of Multiple Deprivation quintile. MMR, measles, mumps and rubella; OR, odds ratio; CI, confidence interval.

found that children from black and mixed/other ethnic backgrounds were less likely to receive timely vaccination, broadly consistent with findings from studies of COVID-19 vaccination uptake which have shown lowest uptake among people from black ethnic backgrounds.[32 33] Additional qualitative evidence suggests women from minority ethnic backgrounds were more likely to find it difficult to access and felt less safe accessing vaccinations for their babies during the COVID-19 pandemic.[34]

Relationships between socioeconomic deprivation and vaccination timeliness appear to be more consistent. A study in Scotland examining vaccination inequity and timeliness demonstrates that the most deprived decile experienced a nearly 50% increased risk of delayed vaccination relative to the least deprived decile for both doses of MMR in the years leading up to the pandemic.[35] Despite this, and in contrast to England, vaccination uptake rose significantly across the first lockdown period in Scotland, with 7000 more children receiving timely routine vaccinations compared with the previous year.[36] The authors of this study speculate that greater flexibility in working patterns offered to many parents during the lockdown period may have increased the accessibility of vaccination appointments.

Our findings are consistent with existing evidence based on Cover of Vaccination Evaluated Rapidly Programme data which confirm that London has a longstanding and disproportionately lower MMR uptake relative to the rest of the UK.[3 37] There is recognised variation between different CCGs in NEL, with lowest uptake in City & Hackney. While particular cultural beliefs held by the Charedi Jewish population in City & Hackney are known to influence uptake of the MMR vaccination,[38] recent evidence suggests that difficulties in accessing vaccination services are also an important factor in this community.[39] The factors responsible for differences between the other CCGs merit further investigation.

Mapping measles vaccinations and outbreaks geospatially enables more granular identification of neighbourhoods requiring focused interventions.[40] Our choropleth maps demonstrate clustering of delayed MMR vaccination in more deprived neighbourhoods—these findings align with previous studies mapping measles outbreak susceptibility[5] and underscore the importance of actionable

real-time information on vaccine timeliness to avert further outbreaks of measles. Our geographical analyses identified an increase in the proportion of children receiving a timely MMR vaccination during the pandemic period in almost one-third of NEL LSOAs. This finding may reflect the innovative measures implemented in some London GPs throughout the pandemic, including vaccinating outside of practice buildings and drive-through services, which may have made routine vaccination more accessible to families.[41]

Our study did not identify an increase in inequity during the pandemic by an area-level measure of deprivation. This is in contrast to findings from Michigan, USA, where the difference in proportion of Medicaid-enrolled children with up-to-date vaccination coverage compared with children not enrolled in Medicaid with up-to-date vaccination coverage increased during the pandemic.[12] This difference is likely to reflect differences in UK and US healthcare systems, as well as the use of an area-level indicator of derivation compared with an individual-level indicator in an area of London with high levels of area-level deprivation.

### Implications for research, policy and practice

Gaps in MMR vaccination coverage increase measles susceptibility, and in 2019, there were 800 000 confirmed cases of measles globally.[42] Measles outbreaks have occurred in 2021 in at least half of the 26 countries that suspended their measles vaccination programmes.[29 42 43] There is evidence that the introduction of social distancing measures, school closures and travel restrictions reduced exposure to vaccine-preventable childhood infections.[44] In an analysis of English hospital admissions, there was a 90% and a 53% reduction in hospital admissions for measles and mumps, respectively, among children aged 0–14 years in the pandemic period compared with the preceding 3-year average, although this study was unable to examine infections managed in primary care.[44] In NEL, we identified 20 measles and 34 mumps cases in primary care during the pandemic period, compared with 325 and 140, respectively, in the equivalent pre-pandemic period.

With a reduction in infection and exposure to infection, measles vaccination may receive less priority in a healthcare system already facing multiple challenges.[45] Awareness and retention of existing WHO targets are critical to prevent measles outbreaks, especially given that measles is the most infectious virus, with a reproduction number of 12–18.[46] The need for targeted public health interventions around routine childhood vaccinations in the context of the pandemic has been recognised internationally,[47 48] as well as in England where a recent campaign by NHS England is encouraging parents of 740 000 children who are not fully vaccinated against MMR to make appointments with their GP.[49] There is strong evidence to support the effectiveness of primary care-led quality improvement programmes to improve vaccine uptake.[37] National measures to tackle these inequalities include NHS England's QOFs to incentivise timely routine childhood vaccinations in primary care.[50] In London, a primary care-led quality improvement programme has been launched to tackle inequalities in timeliness of routine preschool childhood vaccinations.[51]

### CONCLUSION

Routine vaccination schedules have been disrupted during the COVID-19 pandemic. Our study adds important new evidence of the impact on timeliness of MMR vaccinations, and demonstrates unwarranted variation by neighbourhood, ethnicity and deprivation. These data provide further evidence to prioritise quality improvement and catch-up campaigns to achieve herd immunity and prevent measles outbreaks. They provide actionable information in populations and geographies experiencing significant health inequalities.

**Acknowledgements** We are grateful to colleagues within the Clinical Effectiveness Group for access to and expertise in using general practice data. We are also grateful to Professor Bianca de Stavola (Great Ormond Street Institute of Child Health) for her statistical guidance and to Professor Helen Bedford (Great Ormond Street Institute of Child Health) for her clinical expertise and guidance. This work uses data provided by patients and collected by the NHS as part of their care and support. We would also like to acknowledge our PPI participants for their invaluable insights into patient perspectives around dissemination of our research to the wider community.

**Contributors** CD and JR obtained funding for the study. NF and CD conceptualised and designed the analyses. AG, ID and ZA provided guidance for specifying clinical codes. CW and KH extracted data. MM carried out the literature search. NF conducted the analyses, generated tables and figures and drafted the initial manuscript. GH produced the choropleth maps. All authors contributed to the interpretation of analyses and reviewed and revised the manuscript. All authors were involved in writing the paper and had final approval of the submitted and published manuscript. CD is the guarantor and accepts full responsibility for the conduct of the study, had access to the data and controlled the decision to publish.

**Funding** This research was funded by a grant from Barts Charity (ref: MGU0419). This work was supported by Health Data Research UK, which is funded by the UK Medical Research Council, Engineering and Physical Sciences Research Council, Economic and Social Research Council, Department of Health and Social Care (England), Chief Scientist Office of the Scottish Government Health and Social Care Directorates, Health and Social Care Research and Development Division (Welsh Government), Public Health Agency (Northern Ireland), British Heart Foundation and Wellcome. MM is supported by a locally funded National Institute for Health Research Academic Clinical Fellowship.

**Disclaimer** The funders had no role in study design, data collection and analysis, decision to publish or preparation of the manuscript.

**Map disclaimer** The inclusion of any map (including the depiction of any boundaries therein), or of any geographic or locational reference, does not imply the expression of any opinion whatsoever on the part of BMJ concerning the legal status of any country, territory, jurisdiction or area or of its authorities. Any such expression remains solely that of the relevant source and is not endorsed by BMJ. Maps are provided without any warranty of any kind, either express or implied.

**Competing interests** None declared.

**Patient and public involvement** Patients and/or the public were involved in the design, or conduct, or reporting, or dissemination plans of this research. Refer to the Methods section for further details.

**Patient consent for publication** Not required.

**Ethics approval** This study was approved by the Discovery board for service evaluation (measuring what standard of care this service achieved) and analysed routinely acquired de-identified data; hence, no research ethics committee approval was required by the Health Research Authority.

**Provenance and peer review** Not commissioned; externally peer reviewed.

**ORCID iDs**
Nicola Firman http://orcid.org/0000-0001-5213-5044
Milena Marszalek http://orcid.org/0000-0001-5825-0609

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
