## [Reviewer comments · BMJ Open]

ARTICLE DETAILS

TITLE (PROVISIONAL)	Impact of the COVID-19 pandemic on timeliness and equity of measles, mumps and rubella vaccinations in north-east London: a longitudinal study using electronic health records
AUTHORS	Firman, Nicola; Marszalek, Milena; Gutierrez, Ana; Homer, Kate; Williams, Crystal; Harper, Gill; Dostal, Isabel; Ahmed, Zaheer; Robson, John; Dezateux, Carol

VERSION 1 – REVIEW

REVIEWER	Nick Spencer University of Warwick, School of Health and Social Studies
REVIEW RETURNED	17-Sep-2022

GENERAL COMMENTS	This study, based on routinely collected administrative data, is methodologically robust and is an important contribution to the literature on the impact of the C-19 pandemic on routine childhood vaccination coverage. The study focus on MMR coverage is appropriate in the context of North London where MMR coverage rates were below the WHO-recommended of 95%. MMR coverage can also be viewed as a marker for coverage of all routine childhood vaccines. The results confirm the reduction in MMR coverage during the pandemic compared with the pre-pandemic and demonstrate sociodemographic inequalities in coverage by ethnic group and area deprivation. These results are consistent with the findings of our recently published systematic review of the impact of the pandemic on inequity in routine vaccination coverage (reference below). This study found no increase in inequity during the pandemic by area deprivation. By contrast, 4 out of 6 studies included in our review that reported on inequity before and during the pandemic showed increased inequity. I have no suggested changes to the methods or results of this well-designed study; however, I think the authors should cite the recently published systematic review and discuss their findings in the light of its conclusions. The authors cite our initial review (reference 7) which failed to find any studies of routine childhood vaccination coverage inequity during pandemics but the findings of that review have been superseded by the Vaccines paper. Reference: Spencer, N.; Markham, W.; Johnson, S.; Arpin, E.; Nathawad, R.; Gunnlaugsson, G.; Homaira, N.; Rubio, M.L.M.; Trujillo, C.J. The Impact of COVID-19 Pandemic on Inequity in Routine Childhood Vaccination Coverage: A Systematic Review. Vaccines 2022, 10, 1013. https://doi.org/10.3390/vaccines10071013
---

REVIEWER	Helen Skirrow Imperial College London, Public Health
REVIEW RETURNED	20-Sep-2022

GENERAL COMMENTS	Impact of the COVID-19 pandemic on timeliness and equity of measles, mumps and rubella vaccinations in north east London: a longitudinal study using electronic health records Overall comments  - This is a well written article that offers useful information on the characteristics of children who may have had delayed vaccinations during the pandemic and the impact on uptake inequalities. Given London's historical lower uptake of MMR it is a useful geographical cohort. - There are a few elements that need clarification. - There is perhaps slight confusion around the type of work being presented. The study protocol in supplementary file 1 suggests this is an approved research study (as does the STROBE checklist). However, in the ethics statement within supplementary file 1 it states ethics is not required as it is for 'audit' purposes. In the main paper it is described as a service evaluation and correctly states that no ethical approval is required as it is using de-identified data for secondary purposes. In previous service evaluations published it is usual for the service evaluation approval details/registration details to be provided – even service evaluations usually require approval. Maybe add in details of the service evaluation approval by the Discovery Programme board to the main paper ethics statement for full transparency. Audits should not normally be published so need to be clear this is approved SE work given the mention of audit in the protocol provided? Abstracts  - The abstract is well written and concise. Background  - The research question is clear and well presented. - In the background section need to cite the Macdonald paper which reported the initial drop in MMR rates in England in April 2020: McDonald et al https://doi.org/10.2807/1560-7917.ES.2020.25.19.2000848 It is in the study protocol but not the main paper. - The McDonald paper also links to another point: the paper may benefit from adding in some context about what happened to uptake over the course of the pandemic nationally in the UK – in that there were initial steeper declines in the months following the first national lockdown and then some recovery (though overall uptake was lower annually as correctly cited). This could be added to the discussion section where the initial declines at the height of the first wave of the pandemic are discussed in the 'comparison with other literature' section. Methods  - Add a reference to line 109. 'National Health Service (NHS) information governance requirements.' - Figure S2: This sentence is confusing: 'Of 1,882,515 MMR records,
--

	516,358 records were removed as they were not relating to MMR procedures'. What were the 516,358 events related to then? The figure and explanation of the data extraction suggests that the MMR data file was all MMR SNOMED coded events (as per table S1 codes) so why were 516,358 not considered MMR events? Is this actually the '531,469' children with an MMR procedure were not matched to the study denominator as these children are not accounted for in Figure S2? Please clarify. Results  - Results are clear and well presented. - I have not done a formal statistical review however the methods and results seem sound. - Please report on supplementary file 3 table S6 in the results section and not just in the implications section of the discussion. Also not mentioned in the methods section – even though secondary to main outcomes of interest best practice maybe to mention. Discussion  - The conclusions reached are supported by the results. - It is noteworthy that 30.5% of LSOA increased the proportion of children with timely MMR vaccination and maybe worth drawing out in the discussion. See this reference which maybe of interest: https://doi.org/10.3399/BJGPO.2021.0021 - May also be worth considering adding in this reference maybe after line 240 when Scotland is referenced – UK nations experienced different impacts of the pandemic on routine childhood vaccine uptake: https://doi.org/10.1371/journal.pmed.1003916 - This paper may also add some value to discussion around uptake inequalities exacerbated by the pandemic: https://doi.org/10.1016/j.vaccine.2022.06.076 - There is no mention of MMR dose 2 uptake – two doses are essential for protection and it is often this second dose that is missed particularly in more deprived areas. Mention this as a limitation perhaps.
--	--

VERSION 1 – AUTHOR RESPONSE

Reviewer 1			
1	The results confirm the reduction in MMR coverage during the pandemic compared with the pre-pandemic and demonstrate sociodemographic inequalities in coverage by ethnic group and area deprivation. These results are consistent with the findings of our recently published systematic review of the impact of the pandemic on inequity in routine vaccination coverage (reference below). This study found no increase in inequity during the pandemic by area deprivation. By contrast, 4 out of 6 studies included in our review that reported on inequity before and during the pandemic showed increased inequity.	Thank you for bringing to our attention your recently published systematic review which has some very insightful findings which are of great benefit to contextualising the findings of our study. We have updated the introduction to acknowledge this review and highlighted the studies which identified increased inequality in vaccination coverage during the pandemic period.	Introduction
2	I have no suggested changes to the methods or results of this well-designed study; however, I think the authors should cite the recently published systematic review and	Thank you for this suggestion. We have added a paragraph to our discussion comparing	Discussion

	discuss their findings in the light of its conclusions. The authors cite our initial review (reference 7) which failed to find any studies of routine childhood vaccination coverage inequity during pandemics but the findings of that review have been superseded by the Vaccines paper.	our finding of no increased inequity during the pandemic to Bramer et al.'s study.	
Reviewer 2			
1	There is perhaps slight confusion around the type of work being presented. The study protocol in supplementary file 1 suggests this is an approved research study (as does the STROBE checklist). However, in the ethics statement within supplementary file 1 it states ethics is not required as it is for 'audit' purposes. In the main paper it is described as a service evaluation and correctly states that no ethical approval is required as it is using de-identified data for secondary purposes. In previous service evaluations published it is usual for the service evaluation approval details/registration details to be provided – even service evaluations usually require approval. Maybe add in details of the service evaluation approval by the Discovery Programme board to the main paper ethics statement for full transparency. Audits should not normally be published so need to be clear this is approved SE work given the mention of audit in the protocol provided?	Thank you for highlighting this inconsistency. This work is approved by the Discovery board as part of an application for service evaluation to understand at a population level the outcomes and wider impact of COVID-19. We have updated the study protocol (supplementary file 1) and the manuscript to make this clearer. Although this is a service evaluation, we have used the STROBE checklist for robust study reporting.	Ethics approval and supplementary file 1
2	Background: In the background section need to cite the Macdonald paper which reported the initial drop in MMR rates in England in April 2020: McDonald et al https://doi.org/10.2807/1560-7917.ES.2020.25.19.2000848 It is in the study protocol but not the main paper.	Thank you for highlighting this omission. We have now included this citation in the Introduction.	Introduction
3	Background: The McDonald paper also links to another point: the paper may benefit from adding in some context about what happened to uptake over the course of the pandemic nationally in the UK – in that there were initial steeper declines in the months following the first national lockdown and then some recovery (though overall uptake was lower annually as correctly cited). This could be added to the discussion section where the initial declines at the height of the first wave of the pandemic are discussed in the 'comparison with other literature' section.	Thank you for this suggestion, we have commented on the initial decline and short period of recovery as documented in the Public Health England reports.	Discussion
4	Methods: Add a reference to line 109. 'National Health Service (NHS) information governance requirements.'	Thank you for highlighting this omission. A reference has now been provided.	Methods
5	Figure S2: This sentence is confusing: 'Of 1,882,515 MMR records, 516,358 records were removed as they were not relating to MMR procedures'. What were the 516,358 events related to then? The figure and explanation of the data extraction suggests	Thank you for highlighting this area requiring clarification. The data extract containing information about first MMR procedures also included a	Supplementary file 3

	that the MMR data file was all MMR SNOMED coded events (as per table S1 codes) so why were 516,358 not considered MMR events? Is this actually the '531,469' children with an MMR procedure were not matched to the study denominator as these children are not accounted for in Figure S2? Please clarify.	range of other MMR-related events such as vaccination invitations, declines, contraindications, appointment non-attendance and some codes relating to the second MMR vaccination. These events were removed from the data file and only events with a SNOMED clinical code (or those mapped to a SNOMED clinical code if using another coding system) documented in table S2 were retained. We have added this additional information to Table S1 and the footnote under figure S2.	
6	Results: Please report on supplementary file 3 table S6 in the results section and not just in the implications section of the discussion. Also not mentioned in the methods section – even though secondary to main outcomes of interest best practice maybe to mention.	Thank you for this suggestion. We have included the results of this secondary research question in the results section and updated the research questions, methods and supplementary file to reflect this.	Introduction, Methods, Results
7	Discussion: It is noteworthy that 30.5% of LSOA increased the proportion of children with timely MMR vaccination and maybe worth drawing out in the discussion. See this reference which maybe of interest: https://doi.org/10.3399/BJGPO.2021.0021	Thank you for this suggestion. We have reflected on these findings and commented on their role in explaining the increase in vaccination timeliness in some LSOAs in north east London.	Discussion
8	Discussion: May also be worth considering adding in this reference maybe after line 240 when Scotland is referenced – UK nations experienced different impacts of the pandemic on routine childhood vaccine uptake: https://doi.org/10.1371/journal.pmed.1003916	Thank you for this suggestion. We have added a commented on the differences in vaccination uptake and timeliness observed in England and Scotland in the Discussion.	Discussion
9	Discussion: This paper may also add some value to discussion around uptake inequalities exacerbated by the pandemic: https://doi.org/10.1016/j.vaccine.2022.06.076	Thank you for bringing this important qualitative study to our attention. We have reflected on the finds in comparison to those reported in our study in the Discussion.	Discussion
10	Discussion: There is no mention of MMR dose 2 uptake – two doses are essential for protection and it is often this second dose that is missed particularly in more deprived areas. Mention this as a limitation perhaps.	Thank you for this suggestion. We have mentioned this in the Discussion.	Discussion

VERSION 2 – REVIEW

REVIEWER	Nick Spencer University of Warwick, School of Health and Social Studies
REVIEW RETURNED	18-Oct-2022

GENERAL COMMENTS	The authors have responded appropriately to reviewers' comments and I think the paper is now acceptable for publication.
--

REVIEWER	Helen Skirrow Imperial College London, Public Health
REVIEW RETURNED	07-Nov-2022

GENERAL COMMENTS	The comments have been well addressed in this revised manuscript. Interesting paper.
--